# Acute Retroviral Syndrome Presenting with Hemolytic Anemia Induced by G6PD Deficiency

**DOI:** 10.3390/tropicalmed4010006

**Published:** 2018-12-27

**Authors:** Tiago Araujo, Vatsala Katiyar, Jose A. Gonzales Zamora

**Affiliations:** 1John H Stroger Jr Hospital of Cook County, Chicago, IL 60612, USA; vatsala.katiyar@cookcountyhhs.org; 2Department of Medicine, Division of Infectious Diseases, University of Miami, Miller School of Medicine, Miami, Fl 33136, USA; jxg1416@med.miami.edu

**Keywords:** acute HIV, acute retroviral syndrome, G6PD deficiency, hemolysis, hemolytic anemia

## Abstract

Glucose-6-phosphate dehydrogenase (G6PD) deficiency is the most common enzyme defect described in humans. Hemolysis in affected patients is usually triggered by circumstances involving free radical damage. While acute HIV infection is known to be a state of overwhelming oxidative stress, virus-induced hemolytic events in G6PD-deficient patients has rarely been reported. Despite an estimated overall prevalence of 6.8%–13% of this disorder in the HIV population, clinically significant hemolysis has been largely attributed to the use of offending medications rather than HIV infection itself. Here, we present a patient whose first episode of G6PD deficiency-associated hemolysis occurred as the main presentation of acute HIV infection.

## 1. Introduction

G6PD deficiency is an X-linked disorder that leads to hemolysis in the presence of oxidative stress [1]. Prevalence of this condition in the HIV population has been reported between 6.8% and 13% [2,3,4]. While drugs used for treatment and prophylaxis of opportunistic infections in these patients are known to precipitate hemolysis, viral replication by itself has not been attributed as a clinically significant trigger [2,5]. Here, we present a patient who developed G6PD deficiency-induced hemolytic anemia, with acute HIV infection being the precipitating factor.

## 2. Case Presentation

A 22-year-old African-American man presented to the Emergency Department complaining of bright red blood per rectum, diffuse abdominal pain, dark-colored urine, malaise, and 30-pound weight loss in the last month. Bowel habits were unchanged. There was no hematemesis or dysphagia. He also reported vomiting and subjective fevers, but denied dyspnea, cough, night sweats, arthralgia, dysuria, or prior bleeding events. There were no episodes of recurrent infections. His past medical history was unremarkable and he denied taking any medications. He did not report any substance abuse. His sexual history was significant for unprotected same-sex intercourse, with the last encounter two weeks prior to presentation. He reported no recent travels or sick contacts. On physical exam, the patient appeared emaciated and lethargic. There was no pallor, icterus, adenopathy, or rash. Oral examination revealed gingivitis but no thrush or sores. Abdomen was soft and diffusely tender with no distension or guarding. A large, posterior anal fissure was noted on rectal exam with minimal amount of blood. The remainder of examination was unremarkable. No genital ulcers or urethral discharge was noted.

Laboratory studies revealed normal hemoglobin at presentation, but during the course of his hospital stay, he had a significant drop from 14.7 g/dL to 10.3 g/dL over the course of two days (NR 12.9–16.8 g/dL) despite no further bleeding, reaching as low as 8.4 g/dL after the first week. The elevated lactate dehydrogenase of 2100 u/L (NR 85–210 U/L) and slightly increased bilirubin (1.3 mg/dL, NR 0.2–1.2 mg/dL) was suggestive of hemolysis (performed on Beckman Coulter AU 5800). Haptoglobin was found to be low (<6 mg/dL), with a negative direct Coombs test and no schistocytes on the peripheral blood smear. Glucose-6-phosphate dehydrogenase (G6PD) was significantly decreased at 0.4 units/g Hgb (reference range: 4.6 to 13.5 units/gram of hemoglobin). Reticulocyte production index was inappropriately low at 0.48. Vitamin B12 and folate levels were normal, but ferritin was greater than 7500 ng/mL (performed on Beckman Coulter AU 640, NV 23.90–336.20 ng/mL). Platelets were mildly decreased (114 × 103/μL) and white count was within normal limits. He also developed acute kidney injury with creatinine of 1.9 mg/dL upon admission, which resolved two days after fluid resuscitation. Urinalysis revealed a large amount of blood but just one red blood cell per high power field. Urobilinogen was positive and there was also proteinuria 30 mg/dL. Meanwhile he was also worked-up for his abdominal pain with computed tomography (CT) scan, magnetic resonance imaging (MRI), endoscopy, and colonoscopy, all of which were unrevealing.

The patient consented to testing for human immunodeficiency virus (HIV) testing, which revealed positive fourth-generation screening but negative confirmatory results by the western blot technique. Viral load was above the detection limit of two million copies/mL. CD4 lymphocyte count was 456 cells/μL. He was screened for other sexually-transmitted infections with negative hepatitis panel but positive urethral swab for Chlamydia trachomatis and Neisseria gonorrhoeae, for which he received a single dose of ceftriaxone 250 mg and azithromycin 1g. Further infectious work-up that included quantiferon-TB gold, toxoplasmosis serology, and Histoplasma urine antigen were all negative. Epstein–Barr serum DNA and Cytomegalovirus IgM were both negative. The patient received supportive and symptomatic treatment with no blood transfusions. He was started on antiretroviral therapy with dolutegravir, tenofovir alafenamide, and emtricitabine prior to discharge. His hemoglobin remained stable at 7.7 g/dL and his LDH and indirect bilirubin were down-trending after treatment.

## 3. Discussion

The very high HIV RNA levels in our patient is consistent with acute retroviral syndrome. Although he did test positive for HIV screening, the western blot assay was found to be falsely negative. These conflicting results can occur in the first few weeks of infection, as confirmatory tests only detect formed antibodies against the virus. Fourth generation screening, on the other hand, also identifies p24 antigen and thus becomes positive earlier [6].

While anemia is the most common hematological manifestation of HIV infection [7], an acute drop in hemoglobin is not expected. Our patient had several laboratory findings indicative of intravascular hemolysis, including indirect hyperbilirubinemia, elevated lactate dehydrogenase, and low haptoglobin. His inappropriately low reticulocyte production index on days one and three of admission were likely a consequence of suppressed on-demand hematopoiesis. This is seen in HIV infection as erythropoietin levels are depressed and the persistent inflammatory state can muffle its effect on the bone marrow [8,9]. Therefore, the lack of reticulocytosis can be deceptive in this subset of patients undergoing hemolysis.

Various pathophysiological mechanisms can contribute to hemolytic anemia in HIV, such as antibody-mediated, G6PD deficiency and thrombotic microangiopathy [10]. Our patient had a negative direct Coombs test, which is a fairly sensitive marker of autoimmune hemolysis. It should be noted; however, that it can be falsely negative when destruction is mediated by IgA or IgM alone, instead of IgG and C3, which is what the assay is designed to detect [11]. Although thrombotic microangiopathy has been reported as the initial manifestation of acute HIV infection [12], the absence of schistocytes on the peripheral blood smear made this diagnosis unlikely.

Our patient had a very low G6PD level, which was the most likely explanation for his hemolytic anemia. This has been described in a previous case report, in which an African male patient similarly presented with constitutional symptoms and had an acute hemoglobin drop with indirect bilirubinemia, low haptoglobin, and elevated LDH. He was found to have lymphopenia with low CD4 count, which raised concern for HIV infection and, ultimately, revealed a viral load greater than 50 million copies/ml. Extensive laboratory work-up for other causes of hemolysis was unremarkable but he did have low G6PD levels. It was then hypothesized his hemolytic crisis was triggered by a primary HIV infection [13]. This is different from our report in the sense that their patient also had a chronic, low-replicating hepatitis B infection and a positive quantiferon-GOLD. As both mycobacterium and hepatitis B virus can cause hemolytic anemia, it cannot be entirely ruled out that one of these other conditions were not responsible for the hemolysis. Our patient, on the other hand, tested positive for chlamydia and gonorrhea, which have not been associated as triggers for G6PD hemolytic crisis. He received antibiotic treatment after his hemoglobin had already dropped, ruling out a drug-induced etiology. Other important causes of hemolysis that should be excluded include thrombotic microangiopathies, transfusion reactions, disseminated intravascular coagulation, sepsis, malaria infection, and sickle cell disease. Common triggers for hemolysis in patients with G6PD deficiency are listed on Table 1.

The prevalence of G6PD deficiency in the HIV-infected population has been reported to be between 6.8% and 13% [2,3,4]. In these patients, hemolysis is often triggered by medications such as sulfamethoxazole-trimethoprim, isoniazid, dapsone, and primaquine [14]. This is particularly important because these are commonly prescribed drugs for either treatment or prophylaxis of opportunistic infections. In fact, some studies have suggested screening for G6PD deficiency prior to starting these offending agents in patients with susceptible ethnic backgrounds [2]. Fortunately, antiretroviral therapy has not been implicated as a cause of hemolysis [15].

Individuals with G6PD deficiency are more susceptible to red blood cell breakdown in the setting of infection. This is because this enzyme helps to maintain adequate levels of glutathione, which protects cells from free radical damage [1]. Oxidative stress has been reported in asymptomatic HIV-infected patients early in the course of the disease, and reduced levels of glutathione has been found in tissue analysis of these patients [16]. However, the clinical significance of these findings is unclear, since there are studies suggesting that overall survival and number of hemolytic episodes seem to be unaffected by virus-induced oxidative stress [5]. 

Diagnosis can be confirmed by quantification of G6PD, which can be falsely elevated during a hemolytic event, as there is an increase in immature erythrocytes with higher levels of enzyme activity [14]. The decreased levels despite ongoing hemolysis, as illustrated in this case, further reinforces evidence of severe deficiency. In the absence of medications or any other inciting events, acute retroviral syndrome was the most likely precipitating factor for hemolysis in our G6PD-deficient patient.

## 4. Conclusions

Hemolysis from G6PD deficiency can happen as a result of acute HIV infection and should be suspected when there is an acute hemoglobin drop, especially in patients with susceptible ethnic backgrounds. Further data from cohorts and systematic reviews are needed to establish the true frequency and correlation of these events. Physicians should be careful when prescribing medications for opportunistic infections that could precipitate hemolysis in HIV patients and consider screening beforehand.

## Figures and Tables

**Table 1 tropicalmed-04-00006-t001:** Common triggers for hemolysis in G6PD deficiency [1].

Infections	Drugs	Drugs
Hepatitis A	Definitive Association	Possible Associations
Hepatitis B	Sulfamethoxazole	Chloroquine
Cytomegalovirus	Dapsone	Sulfasalazine
Pneumonia	Nitrofurantoin	Aspirin
Typhoid fever	Primaquine	Glibenclamide
	Phenazopyridine	Ciprofloxacin
	Co-trimoxazole	Chloramphenicol
**Food**		Ascorbic acid
Fava Beans		Vitamin K analogues
Bitter melon		Mesalazine

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
