# Peer review of "Acute Retroviral Syndrome Presenting with Hemolytic Anemia Induced by G6PD Deficiency"

_tropicalmed, 2018, doi:10.3390/tropicalmed4010006_

Round 1
Reviewer 1 Report
This is a nice, straightforward case study. The writing has a good flow; there are just a few tweaks required. See below for a list of suggested changes.
Lines 13-14: I don’t think that you can unequivocally state that the hemolysis was the main presentation of acute HIV as the patient initially presented with lethargy, history of vomiting and possible fevers, all of which are clinical signs that can be associated with acute HIV, the hemolytic event occurred soon after presentation
-there is a history of dark coloured urine at presentation, what was this attributed to? What were the urinalysis results?
Line 28: change “endorsed” to “reported”
Line 36: change ”Rest of” to “The remainder of”
Line 39:-add in the normal range for Hb, LDH and indirect bilirubin at your laboratory, also add in at what day in the hospital stay that the drop in Hb occurred and how long it took for it to occur (was it within a day or over a few days??)
Line 40: many bilirubin tests can have false positives due to interference by hemolysis, can this occur with the test that your institution uses? Add in the total bilirubin values and normal interval, also add in the specific tests and automated machines that were used to perform the measurements
Line 41: change ”suggested” to “suggesting”
Line 45: add in the normal reference interval for ferritin and the testing method used
Line 50: change ”We consented the patient to” to: “The patient consented to testing for…‘
Line 62” change ”notoriously” to “very”
Line 62: add ”the” before western
Line 70: how long did it take for the Hb to drop in this patient, if it was very acute, then the lack of reticulocytosis could also be attributed to insufficient time for the bone marrow to respond, however if the patient was in the hospital for several days following the development of anemia and still didn’t respond, then there is support for the theory that the bone marrow was unable to respond, please add in more to support you hypothesis or add in that the lack of reticulocytosis could be due to no time for a response
Line 88: isn’t clear whether “this patient” refers to the patient in this case report or the patient that was previously described, please be more specific
Line 89: presuming that the +ve GOLD test etc describes the patient in the other case study… add in “as both mycobacterium and hepatitis B can cause a hemolytic anemia, therefore it cannot be ruled out in that case that one of these other conditions was responsible for the hemolysis”.
Author Response
Lines 13-14: I don’t think that you can unequivocally state that the hemolysis was the main presentation of acute HIV as the patient initially presented with lethargy, history of vomiting and possible fevers, all of which are clinical signs that can be associated with acute HIV, the hemolytic event occurred soon after presentation
Answer
I see your point. Our thought process was that most of these symptoms were being caused by hemolysis itself, although they are indeed nonspecific and could be attributed to either hemolysis or HIV infection. The patient did complain of dark-colored urine from the beginning (and I remember looking at the urine at bedside and seeing a very typical cola-like appearance) and we don't have a baseline of Hgb prior to the initial admission to rule in/out hemolysis already happening (example: his baseline was 16 and when he got to us it was 14). In any case hemolysis happened very early on the course of the infection, so maybe a change in wording will suffice?
There is a history of dark coloured urine at presentation, what was this attributed to? What were the urinalysis results?
Answer
UA results: protein-30, glucose – neg, blood-large, RBC -1, WBC-3. Bilirubin negative. Urobilinogen 4 (Reference value is negative)
(large blood with less RBC can happen in Hburia and myoglobinuria)
Line 39:- add in the normal range for Hb, LDH and indirect bilirubin at your laboratory, also add in at what day in the hospital stay that the drop in Hb occurred and how long it took for it to occur (was it within a day or over a few days??)
Answer
Reference values are Hb: 12.9-16.8 g/dl . LDH: 85-210 U/L
Indirect bilirubin is calculated by subtracting total bilirubin and direct bilirubin, so no normal values
Hb on D1- 14
Hb on D10 – 8.3, but maximum HB drop occurred from D1 to D3 (13.9 to 10.3).
Line 40: many bilirubin tests can have false positives due to interference by hemolysis, can this occur with the test that your institution uses? Add in the total bilirubin values and normal interval, also add in the specific tests and automated machines that were used to perform the measurements
Answer
No, bilirubin tests in our laboratory are not affected by hemolysis to the best of technician’s knowledge. Automated machine used in our lab: Beckman coulter AU 5800
Reference values
Total bili: (0.2-1.2 mg/dl)
Direct bili: (0.0-0.2 mg /dl)
So according to calculations, normal indirect bili would be 0.2-1 mg/dl
Line 45: add in the normal reference interval for ferritin and the testing method used
Answer
23.90-336.20 ng/ml
Method: ferritin test immunoassay kit and then after the kit is used, Beckman coulter is used to determine the values
Line 70: how long did it take for the Hb to drop in this patient, if it was very acute, then the lack of reticulocytosis could also be attributed to insufficient time for the bone marrow to respond, however if the patient was in the hospital for several days following the development of anemia and still didn’t respond, then there is support for the theory that the bone marrow was unable to respond, please add in more to support you hypothesis or add in that the lack of reticulocytosis could be due to no time for a response
Answer
23.90-336.20 ng/ml
As mentioned Hb dropped max over 2 days. Upon presentation retic was 0.5 (on 8/9/2017) and another retic count on 8/11/2017 was 1.4 (normal: 0.3-2.7). We don’t have any more repeats. Even though his Hb continued to drop, he did not have an appropriate response
Reviewer 2 Report
The manuscript could be improved by adding information about other putative causes of increased hemolysis e.g. hemoglobin composition (sickle cell disease?) and pyruvate kinase activity
Author Response
Thank you for the suggestion, we will try to briefly add a few causes of increased hemolysis to the discussion.
Reviewer 3 Report
This paper is the presentation of a case report regarding the haemolytic anemia in a G6PD deficient patient during an acute retroviral syndrome. The coexsistence of HIV infection and G6PD deficiency is not rare being the reported prevalence in HIV population between 6.8 and 13%. In these patients many different infections are generally found as in this patient who was positive for Clamydias trachomatis and Neisseria gonorrhorreae and for these infections the treatment with a single dose of ceftriaxone 250 mg and azithromycin 1 g was applied.
The Authors reported a very low G6PD level as the most likely explanation for the haemolytic anaemia, but the patient has received drugs and so the Authors should specify better the following points:
1. if the haemolysis during the hospital stay was observed before or after the drugs therapy for Clamydias trachomatis and Neisseria gonorrhorreae ;
3. the effect of the antiretroviral therapy on the anemic state of the patient;
3. a comment regarding why the haemoglion level remains low at 7.7 g/dL.
Author Response
1. if the haemolysis during the hospital stay was observed before or after the drugs therapy for Clamydias trachomatis and Neisseria gonorrhorreae ;
Answer
Azithro and ceftriaxone were on 8/22 . Hb was stable at 7.7 by that time. So it further supports our theory
3. the effect of the antiretroviral therapy on the anemic state of the patient;
Answer
Unfortunately, the patient moved to another state and has very poor follow-up there. We asked him the name of the place where he was getting his care and called the clinic but they refused to give us information, the patient needs to request release of records himself.
3. a comment regarding why the haemoglion level remains low at 7.7 g/dL.
Answer
It was the Hb on discharge, shortly after being started on HAART. Patient further did not follow up at our hospital